

# Skin and fur bacterial diversity and community structure on American southwestern bats: effects of habitat, geography and bat traits

Ara S. Winter[1], Jennifer J.M. Hathaway[1], Jason C. Kimble[1], Debbie C. Buecher[2], Ernest W. Valdez[1,3], Andrea Porras-Alfaro[4], Jesse M. Young[1], Kaitlyn J.H. Read[1] and Diana E. Northup[1]

[1] Department of Biology, University of New Mexico, Albuquerque, NM, USA
[2] Buecher Biological Consulting, Tucson, AZ, USA
[3] Fort Collins Science Center, U.S. Geological Survey, Fort Collins, CO, USA
[4] Department of Biological Sciences, Western Illinois University, Macomb, IL, USA

Corresponding author
Ara S. Winter,
ghashsnaga@gmail.com

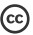

## ABSTRACT

Microorganisms that reside on and in mammals, such as bats, have the potential to influence their host's health and to provide defenses against invading pathogens. However, we have little understanding of the skin and fur bacterial microbiota on bats, or factors that influence the structure of these communities. The southwestern United States offers excellent sites for the study of external bat bacterial microbiota due to the diversity of bat species, the variety of abiotic and biotic factors that may govern bat bacterial microbiota communities, and the lack of the newly emergent fungal disease in bats, white-nose syndrome (WNS), in the southwest. To test these variables, we used 16S rRNA gene 454 pyrosequencing from swabs of external skin and fur surfaces from 163 bats from 13 species sampled from southeastern New Mexico to northwestern Arizona. Community similarity patterns, random forest models, and generalized linear mixed-effects models show that factors such as location (e.g., cave-caught versus surface-netted) and ecoregion are major contributors to the structure of bacterial communities on bats. Bats caught in caves had a distinct microbial community compared to those that were netted on the surface. Our results provide a first insight into the distribution of skin and fur bat bacteria in the WNS-free environment of New Mexico and Arizona. More importantly, it provides a baseline of bat external microbiota that can be explored for potential natural defenses against pathogens.

Subjects Biogeography, Bioinformatics, Ecology, Microbiology
Keywords Bat, Microbiome, 16S rRNA, Chiroptera, White-nose syndrome, WNS, Skin microbiome, Bat microbiota

## INTRODUCTION

Recent studies of microbiomes associated with humans and some other species have shown that external and internal microbiota play critical roles in maintaining the health and well-being of these organisms (*Apprill et al., 2014*; *Brucker et al., 2008*; *Harris et al., 2009*; *Human Microbiome Project Consortium, 2012*; *Lowrey et al., 2015*). In contrast to

humans, the nature of the microbiota associated with bats, in particular with their skin and fur surfaces, is poorly studied. Furthermore, we know very little about what role bacteria play in defense against invading pathogenic microorganisms in bats, a diverse group of mammals that plays key roles in our agriculture and natural ecosystems (*Boyles et al., 2011*).

In humans, factors such as age, sex, and geography all play a significant role in shaping a person's microbiota (*Ying et al., 2015*). Outside of humans, skin microbiomes have received less attention than their counterparts in the gut. However, a few studies have highlighted the diversity of skin microbiota in other animals. For example, on rainbow trout, the different regions of the skin showed diverse mucosal surface microbiota, which were dominated by Proteobacteria (*Lowrey et al., 2015*). Other studies of mammals have highlighted the importance of location and habitat on diversity of skin microbiota. *Apprill et al. (2014)* found humpback whales from different oceans possessed significantly different microbiota on their skin, thus indicating that geography and local habitat are indicators of regional types of skin microbiota. This geographic difference is also observed in Tasmanian devils where there is a distinction between captive and wild devils in terms of their skin microbiota (*Cheng et al., 2015*). Recently, two external microbiome studies by *Lemieux-Labonté et al. (2016)* and *Avena et al. (2016)* on neotropical and temperate bats, respectively, have shown that the external microbiota differed mostly by habitat, with species also influencing the microbiota.

The role of skin microbiota in the overall health of an animal is just beginning to be understood. Therefore, in addition to being diverse across geographic locations, skin microbiota may also play a role in susceptibility to disease. Changes in the human skin microbiota have been observed in such diseases as atopic dermatitis and psoriasis (*Kong et al., 2012*; *Statnikov et al., 2013*). The nature of the complex interactions of skin microbiota is still poorly understood, but is thought to play a crucial role in an animal's ability to fight off disease (*Belden & Harris, 2007*). For example, isolates from the skin of rainbow trout were shown to inhibit the growth of two pathogenic fungi (*Lowrey et al., 2015*). In salamanders, the presence of bacterial symbiont *Janthinobacterium lividum* can inhibit the skin pathogen *Batrachochytrium dendrobatidis*, a deadly fungal infection that has caused widespread devastation to salamander and frog populations (*Brucker et al., 2008*).

Recent interest in the external microbiota of bats has increased due to the effects of white-nose syndrome (WNS), which was introduced into the eastern region of the United States approximately 10 years ago (*Frick et al., 2010*). WNS is caused by a psychrophilic, keratinophilic fungus (*Pseudogymnoascus destructans*) that attacks bat wings and uropatagium (tail membrane) during hibernation. Besides causing excessive arousals during hibernation, it leads to degradation of the physiological function of a large surface area on the bats, as well as disruption to fat storage and water regulation. Currently, WNS has killed millions of hibernating bats in eastern North America and is spreading westward. The impacts from this wildlife disease have the potential to affect multiple species of bats across North America. Second to rodents, bats are the most species rich mammal order in the world (*Wilson & Reeder, 2005*). Chiroptera is represented by

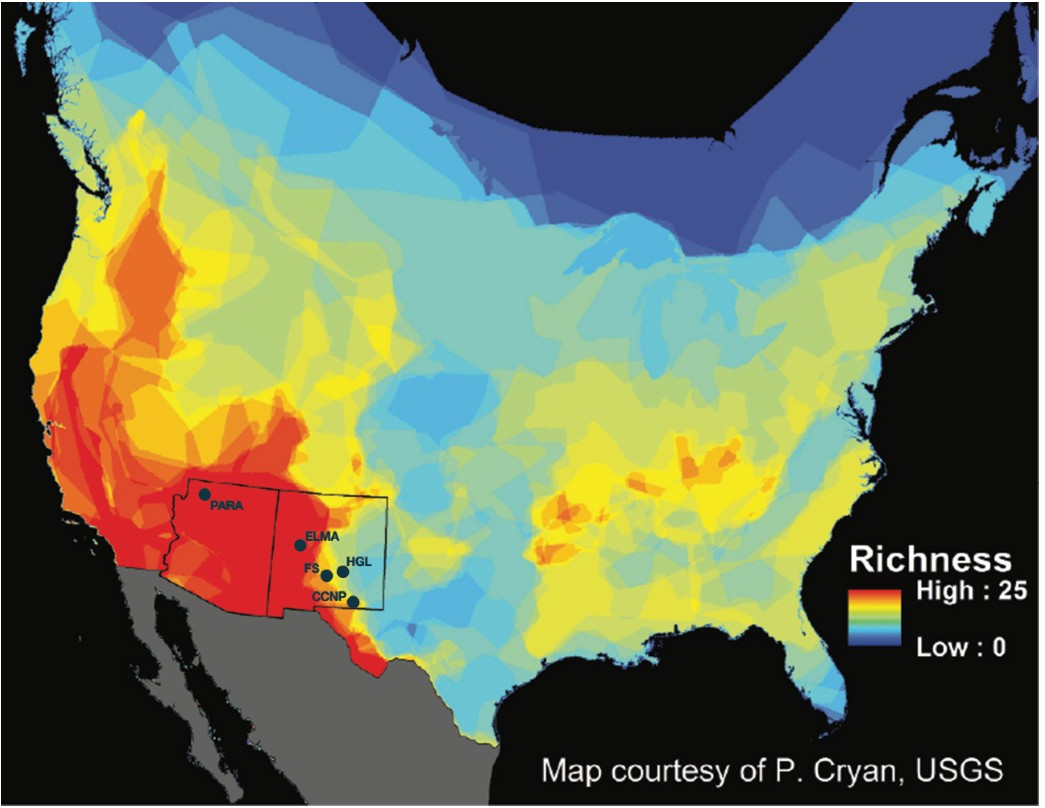

**Figure 1** **New Mexico and Arizona are hotspots of bat diversity.** Map of bat species richness in the United States (US) and Canada. Total number of bat species occurring in each area calculated by counting the number of overlapping species distributions. Warmer colors represent areas with higher species richness and cooler colors represent areas with lower species richness. Sample locations are marked as follows: PARA (Grand Canyon-Parashant National Monument), ELMA (El Malpais National Monument), FS (Fort Stanton-Snowy River Cave National Conservation Area), HGL (High Grasslands), CCNP (Carlsbad Caverns National Park). Map courtesy of P. Cryan, USGS.

approximately 1,116 different species that occupy habitats ranging from the wet tropics of the equator to dry lowland deserts in temperate latitudes (*Wilson & Reeder, 2005*). Within the continental United States, there are approximately 45 species of bats, of which 28 species occur within the Southwest, belonging to Vespertilionidae, Molossidae, Phyllostomidae, and Mormoopidae. Many of these bat species are sympatric and syntopic, especially in New Mexico and Arizona (*Findley et al., 1975*; *Humphrey, 1975*; *Hall, 1981*; *Hoffmeister, 1986*; *Frey, 2004*; *Harvey, Altenbach & Best, 2011*) (Fig. 1).

   Given the high diversity of bat species in the western and southwestern United States, the potential threat to bat diversity at a regional-scale is very high. Arizona and New Mexico have nine species of *Myotis*, some of which are western analogs to eastern bat species currently impacted by WNS. It is therefore critical that we determine which western species will be negatively impacted by WNS prior to its predicted arrival (*Maher et al., 2012*) in order to target our monitoring for WNS. It is possible that certain bacteria present on some bat species can influence the progression and outcome of WNS (*Hoyt et al., 2015*).

Because *P. destructans* is a novel species for cave ecosystems in North America, it is likely affecting the natural external microbiome of bats and caves. Aside from the study by *Avena et al. (2016)*, knowledge on a regional-scale of the external bat microbiome in a WNS-free area is lacking, particularly in the southwest where species diversity of hibernating bats that can be affected by WNS is greatest. Local factors, including abiotic and biotic variables in geographic patterns of the bat external microbiome at the local and regional-scale, are needed in order to understand the potential natural defenses of the external bat microbiota.

In this study, we analyzed samples from 163 bats collected from southeastern New Mexico to northwestern Arizona to gain insights into regional-scale patterns of skin and fur bat bacteria and the factors that drive these patterns. Specifically, we address two questions: First, does the daily routine of bats influence species richness or taxonomic distribution of bacterial composition on skin and fur; Second, to what extent are the changes in distributions of bacteria on bat skin and fur a function of habitat (across varying spatial scales), host species, and host behavior? In this study, "microbiota" is used to mean the 16S rRNA gene survey that was used to taxonomically identify bacteria on bat skin and fur.

## METHODS

### Sampling

We initially sampled 186 bats belonging to 13 species (*Myotis ciliolabrum*, *Myotis californicus*, *Myotis evotis*, *Myotis occultus*, *Myotis thysanodes*, *Myotis velifer*, *Myotis volans*, *Corynorhinus townsendii*, *Eptesicus fuscus*, *Tadarida brasiliensis*, *Antrozous pallidus*, *Parastrellus hesperus*, and *Lasionycteris noctivagans*) using 16S rRNA gene analysis for skin and fur microbiota identification. These samples came from five study locations in the southwest: Grand Canyon-Parashant National Monument (PARA) in Arizona, and Carlsbad Caverns National Park (CCNP), Fort Stanton-Snowy River Cave National Conservation Area (FS), El Malpais National Monument (ELMA), and Bureau of Land Management high grasslands (HGL) caves near Roswell, in New Mexico (Fig. 1). Bat sample collection was allowed under the following permits: 2014 Arizona and New Mexico Game and Fish Department Scientific Collecting Permit (SP670210, SCI#3423, SCI#3350), National Park Service Scientific Collecting Permit (CAVE-2014-SCI-0012, ELMA-2013-SCI-0005, ELMA-2014-SCI-0001, PARA-2012-SCI-0003; BLM LLNMP0RF0-14-0504001 and LLNMP01400-13-0920), Fort Collins Science Center Standard Operating Procedure (SOP) SOP#: 2013-01, and an Institutional Animal Care and Use Committee (IACUC) Permit from the University of New Mexico (Protocol #15-101307-MC) and from the National Park Service (Protocol #IMR_ELMA.PARA.CAVE.SEAZ_Northup_Bats_2015.A2). Figure S1 summarizes the bat species distribution across ecoregions.

Samples were collected between spring and early autumn from 2012 through 2014. Cave-caught bats were either plucked from the walls of the caves (Fig. 2a) in ELMA, FS, and HGL, or netted in sterilized nets in Carlsbad Cavern in CCNP in a location along their flight path out of the cave. Cave-caught bats were typically sampled 6–8 h after returning
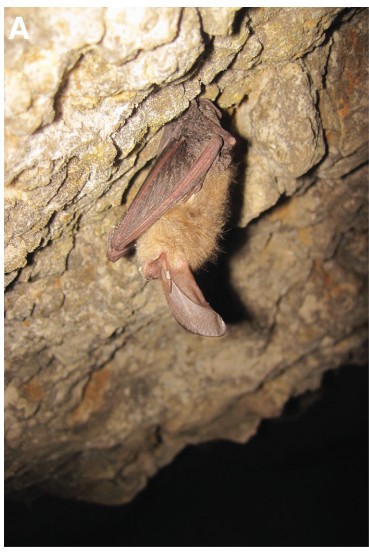 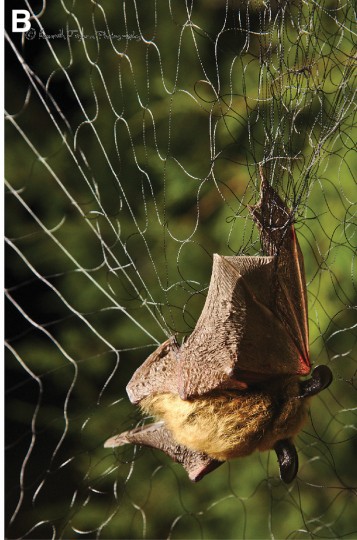

**Figure 2 Differences in captures: cave and surface captures.** (A). Townsend's big eared bat (*Corynorhinus townsendii*) roosting on a cave wall, with stomach and uropatagium in contact with wall. (B) Long-eared bat (*Myotis evotis*) captured in a surface mist net. Photographs by Kenneth Ingram.

to the cave in the early morning. Surface-netted bats were netted after sundown using sterilized nets near water sources in CCNP, ELMA, FS, and PARA.

Surface-netted bats (Fig. 2b) may roost in surface features (trees, rock crevices, old buildings) or have been out of the cave for an indeterminate amount of time prior to capture and swabbing.

All bats were handled with clean gloves and swabbed for DNA before other measurements were taken to limit contamination by human-associated microbiota. Using a sterile swab moistened with Ringer's Solution (*Hille, 1984*), the entire skin (i.e., ears, wings and uropatagia) and furred surfaces of each bat were thoroughly swabbed. While the bat biologist held the bat in appropriate positions to give access to the area to be swabbed, the microbiologist rubbed each area with the sterile swab approximately three to five times, rotating the swab as the action was performed.

Each swab was placed in a sterile 1.7 ml snap-cap microcentrifuge tube containing 100 µl of RNAlater®, and immediately frozen in a liquid nitrogen dry shipper or placed on dry ice. Samples were transported to the University of New Mexico and stored in a −80 °C freezer. Samples were sent to MR DNA Molecular Research LP, Shallowater, Texas (http://www.mrdnalab.com/) for genomic DNA extraction and 454 sequencing diversity assays of bacterial 16S rRNA genes. The 186 samples were sequenced in nine runs. Barcoded amplicon sequencing processes were performed by MR DNA® under the trademark service (bTEFAP®). The 16S rRNA gene (V1–V2) universal PCR primers 27F (5′-AGRGTTTGATCMTGGCTCAG-3′) and 519 R (5′-GWATTACCGCGGCKGCTG-3′) were used in a single-step 30 cycle PCR using the HotStarTaq Plus Master Mix Kit (Qiagen, Valencia, CA, USA) under the following conditions: 94 °C for 3 min, followed by 28 cycles (five cycle used on PCR products) of 94 °C for 30 s, 53 °C for 40 s and

72 °C for 1 min, after which a final elongation step at 72 °C for 5 min was performed. Sequencing with the 27F primer was performed at MR DNA on a Roche 454 FLX titanium following the manufacturer's guidelines.

## CONSIDERATIONS ABOUT THE DATA SET

1. Bats were caught (plucked from the cave wall) or netted (surface mist nets) in natural settings.

2. The whole bat body was swabbed to survey the broad skin and fur bat microbiome because our observations have shown that during roosting in caves much of the fur, rather than the whole wings, is in contact with cave walls and is likely picking up cave bacteria.

3. In this study, there was no way to determine if a surface-netted bat was day roosting in a non-cave surface feature or in a nearby cave.

4. Due to no control samples for this study we cannot rule out the possibility of capturing transient bacteria on the bat skin and fur.

### Sequence processing

All 454 reads were processed in QIIME 1.9 (*Caporaso et al., 2010*). Primer and linker sequences were removed before analysis. Bacterial sequences shorter than 200 bp or longer than 500 bp, or containing bases with a quality score lower than 30, were excluded. The quality control and trimming was computed using the split_libraries command. Bacterial samples were denoised and clustered into operational taxonomic units (OTUs) (at the 97% level) with pick_denovo_otus.py pipeline using the sumaclust option (*Mercier et al., 2013*). Chimera checking was done using usearch (*Edgar, 2010*) to detect artifacts created during sequencing. Taxonomy was assigned using SILVA123 database with uclust. All hits to the SILVA database that were classified as Chloroplast sequences were removed in QIIME with filter_taxa_from_otu_table.py. Full QIIME workflow with all parameters used is available at: https://zenodo.org/record/17577#.

From the initial 186 samples 23 were removed including one bat species. Samples that were removed had either: Missing metadata, unclear sample IDs, low occurrence (less than three bats of the same species caught in a single location), or were filtered out in the sequencing processing steps. The bat species removed was a single *Lasiurus cinereus* that was caught and swabbed.

### Metadata explanation

Complete metadata is included in Table S1. The data for the project were coded with several pieces of metadata to help in the sample tracking and analysis. Bat number is for internal use to match 16S rRNA gene IDs to Buecher's database. Host_species is the bat species sampled for this project. Spec_abbv is the four-letter abbreviation of the bat species. Sex of the bat is determined by the bat biologists in the field. Area is the national park or Bureau of Land Management (BLM) land on which the bats were sampled. Local_habitat is the site where the bats were caught or netted. Cave_or_surface indicate if

the bats were cave-caught or surface-netted. State code is where the bats were sampled: NM for New Mexico and AZ for Arizona. Lat is the latitude of the local habitat. Ecoregion_iv is the EPA Ecoregion IV (*Omernik & Griffith, 2008*) designation for the local habitat. Season is the season in which the bats were sampled. Forearm and mass were measured on the bats in the field. Feeding_Flight_behavior and Diet are the typically behaviors for the bat species. Date, month, and month_cat are numerical dates and categorical data when the bats were sampled.

## Normalizing the data

Microbiome studies deal with differences in library sizes (number of sequences per sample) in a variety of ways. The once standard practice of rarefying data (subsampling to an even depth) is now statistically inadmissible (*McMurdie & Holmes, 2014*) for microbial abundance data. While many important discoveries were made with rarefied data, doing this removes real data (removal of OTUs); removes samples that can be clustered meaningfully by other methods (NMDS, DESeq2); results in loss of statistical power; and increases false positive rates when comparing abundance data across categories (see *McMurdie & Holmes, 2014* for further details). Transformation of the count and richness data was carried out using the normalize_table.py in QIIME, with the cumulative sum scaling (CSS) (*Paulson et al., 2013*) option. CSS divides the raw counts by the sum of counts in a sample, up to a percentile determined by normalize_table.py. This normalizes that data to account for differences in library size for all downstream analysis. Figure S2 shows the rarefaction curves for the raw alpha diversity indices: total richness, Chao1, Chao1 standard error, and Shannon.

## Alpha diversity analysis and normalization

Alpha diversity indices were calculated in QIIME using alpha_diversity.py command. Rarefaction curves plotted against observed species, chao1, chao1 standard error, and Shannon are available in the supplemental data (Table S1).

## Distribution of major bacterial taxa on bats

Cleveland dot plots of major phyla and classes of interest were run in ggplot. Proteobacteria were further divided into the following classes: Alphaproteobacteria, Betaproteobacteria, and Gammaproteobacteria. Each plot is the relative abundance of the phylum and class within a sample. The R script and data for these plots are provided in a link in the data and workflow availability section. The core bat microbiome, as defined as bacteria that are common across 90% of the bats sampled, was determined using compute_core_microbiome.py in QIIME.

## DESeq2—differential abundance between cave-caught and surface-netted bats

DESeq2 (*Love, Huber & Anders, 2014*) was used to identify taxa that were differentially proportional between cave-caught and surface-netted bats and across ecoregions. DESeq2 was picked due to its ability to correct for large differences in sample library size without loss of statistical power or increase in false positive rates. A custom script for running

DESeq2 in R by Umer Zeeshan Ijaz (http://userweb.eng.gla.ac.uk/umer.ijaz, accessed 4th March 2016) was used. Figure S3 shows the differential abundance box plots by cave-caught and surface-netted.

## Classifying bat species and habitats

Random forest models were run in QIIME (supervised_learning.py) using 10-fold cross-validation with 1,000 trees. The random forest models were run to test if our classes of samples were predictive of the bacterial community composition.

## Dissimilarity of bat bacterial taxa by habitat

NMDS analysis was carried out using the phyloseq package (*McMurdie & Holmes, 2014*) and ggplot2 (*Wickham, 2009*) in R 3.3.2 (*R Development Core Team, 2012*). The main analysis was focused on drivers of beta diversity across different categories. NMDS is robust to large differences in count data. The Bray–Curtis distance was used because it is invariant to changes in units and unaffected by additions of new communities, and NMDS was chosen because it uses rank orders and does not assume linear relationships.

## Bat bacterial richness latitudinal gradient

The bayesboot (*Bååth, 2016*) package for R was used to visualize bacterial richness found on bats as it changes across latitudes. Bayesboot is a Bayesian bootstrap package for summary statistics and modeling.

## Modeling bacteria similarity and richness

Modeling of environmental parameters and grouping data were done in R using the rstanarm (*Gabry & Goodrich, 2016*) package with a generalized linear mixed effects model (glmer). We chose a Gaussian family; a normal, weakly informed prior (normal(location = 0, scale = 8)); and 10,000 iterations. Grouping data were treated as random effects in a partial pooling model. The Rhat statistic was used to measure if the MCMC chains converged. Rhat measures the ratio of the average variance of the draws within each chain to the variance of the pooled draws across chains. Full model results are available in Data S1.

## Data and workflow availability

Biome files, QIIME mapping files, workflow, and R scripts are available at https://github.com/bioinfonm/microBat/tree/batmicrobiom and are archived at https://zenodo.org/record/17577#. All raw sequence data with the quality files and mapping files are available at: https://zenodo.org/record/50976. The full metadata table is available in the supplemental data (Table S1). A Binder (http://mybinder.org/) ipython notebook with the full dataset is available at: https://github.com/bioinfonm/microBat.
Cave names and locations are encoded to protect park and BLM resources.
The full cave names and sampling locations are protected by federal law and their respective agencies.

## RESULTS

### Sample statistics

After sequence processing and sample data cleaning, there were a total of 163 bats sampled for the 16S rRNA gene study. Of the 163 bats sampled, 60 were cave-caught and 103 were surface-netted. There were 65 female and 95 male bats sampled. The distribution number of bats sampled by bat species and by ecoregion is shown in Fig. S3. After quality control, the number of reads range from 843 to 20,515 per sample. Sample coverage was measured by calculating the Good's coverage, whose values (Table S1) ranged from 81% to 99%, with an average of 95.3%. After normalization of the data the number of OTUs per sample ranged from 468 to 12,135 with a standard deviation of 2,557.

Very small portions (0–0.25%) of the sequences in the data could not be assigned to a phylum, but were identified as bacterial. At the class level, portions between 0% and 0.55% could not be assigned to a class. Actinobacteria (phylum), Alphaproteobacteria (class), Gammaproteobacteria (class), and Firmicutes (phylum) made up the most abundant taxa across all bat species (Fig. 3).

Cave-caught bats were dominated by the phylum Actinobacteria (Fig. S1), whereas surface-netted bats were dominated by Cyanobacteria, Actinobacteria, and Alphaproteobacteria (Fig. S2). These same phyla are differentially abundant as determined by DESeq2 in cave-caught (plucked from the walls) and surface-netted.

There is a small number of bacteria that are shared across 90% of the bats samples. These bacteria are represented by the following taxa: the Actinobacteria *Micrococcaceae* and *Intrasporangiaceae* and the Bacteroidetes *Flavobacteriaceae*.

### The effect of habitat, ecoregion and species type on skin microbiota

There was variation in the skin/fur bat microbiome composition at different spatial scales (habitats) from cave-caught or surface-netted, EPA ecoregion IV (*Omernik & Griffith, 2008*), to individual sampling sites. However, these data are cofounded with many variables including: the presence of bat species, number of cave-caught or surface-netted bats present, and changing spatial scales. The confounding variables led us to broadly classify the habitats using the random forest model. The goal of the random forest model is to classify unlabeled communities based on a set of labeled training communities. This will generate a ratio of estimated generalization error and baseline error. A reasonable ratio of the estimated generalization error compared to the baseline error should be two or greater, i.e., the random forests classifier does at least twice as well as random guessing for an unlabeled community.

Using the proposed random forest model we tested whether the data could be classified by our metadata categories. The models were successful for determining cave-caught or surface-netted bats with a ratio of 10.1 and ecoregion with a ratio of 3.5. Random forest models were also successful for determining local habitat (3.55) and bat species (2.47) associated with each sample. However, the model was unsuccessful for feeding flight behavior (1.12). Although the bat bacteria could not be categorized by bat feeding/flight behavior in the random forest model, the data are still relevant for overall bacterial

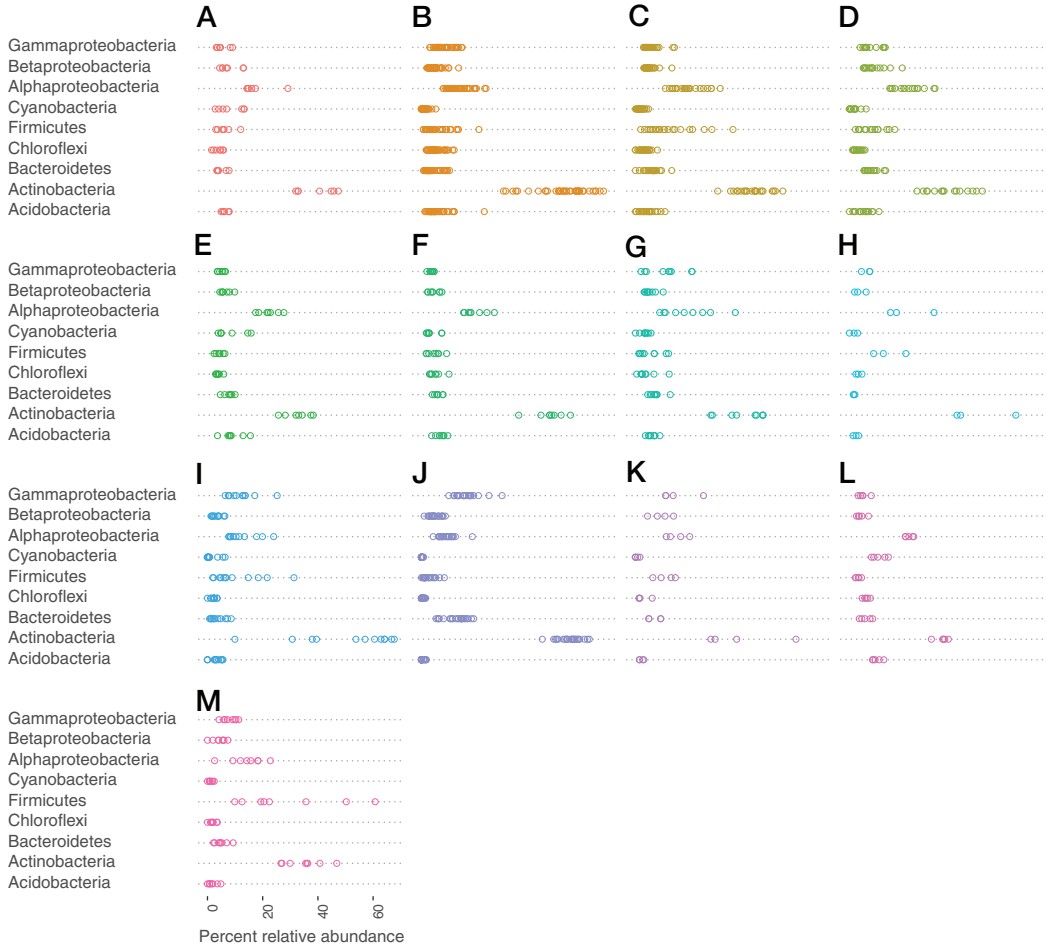

**Figure 3 Bacteria phylum varied by bat species.** Cleveland dotplot of the relative abundance of the top nine bacterial phyla/Proteobacteria class grouped by bat species. Each dot in a phylum category represents a single sample point. (A) *Antrozous pallidus*, (B) *Corynorhinus townsendii*, (C) *Eptesicus fucus*, (D) *Lasionycteris noctivagans*, (E) *Myotis californicus*, (F) *Myotis ciliolabrum*, (G) *Myotis evotis*, (H) *Myotis occultus*, (I) *Myotis thysanodes*, (J) *Myotis velifer*, (K) *Myotis volans*, (L) *Parastrellus hesperus*, (M) *Tadarida brasiliensis*.              

richness and bacterial community dissimilarity. Using the random forest model to take the OTU counts as predictors and the metadata (i.e., ecoregion) as classes, we can classify a given bat microbiome as cave-caught or surface-netted and from which ecoregion it came.

Patterns in community dissimilarity were measured using NMDS (Fig. 4) combined with Bayesian hierarchical models to explain the predictors of dissimilarity. Bats that were cave-caught cluster more closely together than bats that were surface-netted. Bats sampled within an ecoregion ranged from being tightly clustered (Chihuahuan Basins and Playas) with each other (more related within an ecoregion) to highly variable (more related between ecoregions, Lava Malpais). Due to the number of species and sampling habitats these data are not shown for the NMDS.

A Bayesian hierarchical model was fit to explain the amount each predictor contributed to the community dissimilarity on the NMDS1 and NMDS2 axis. All values are reported

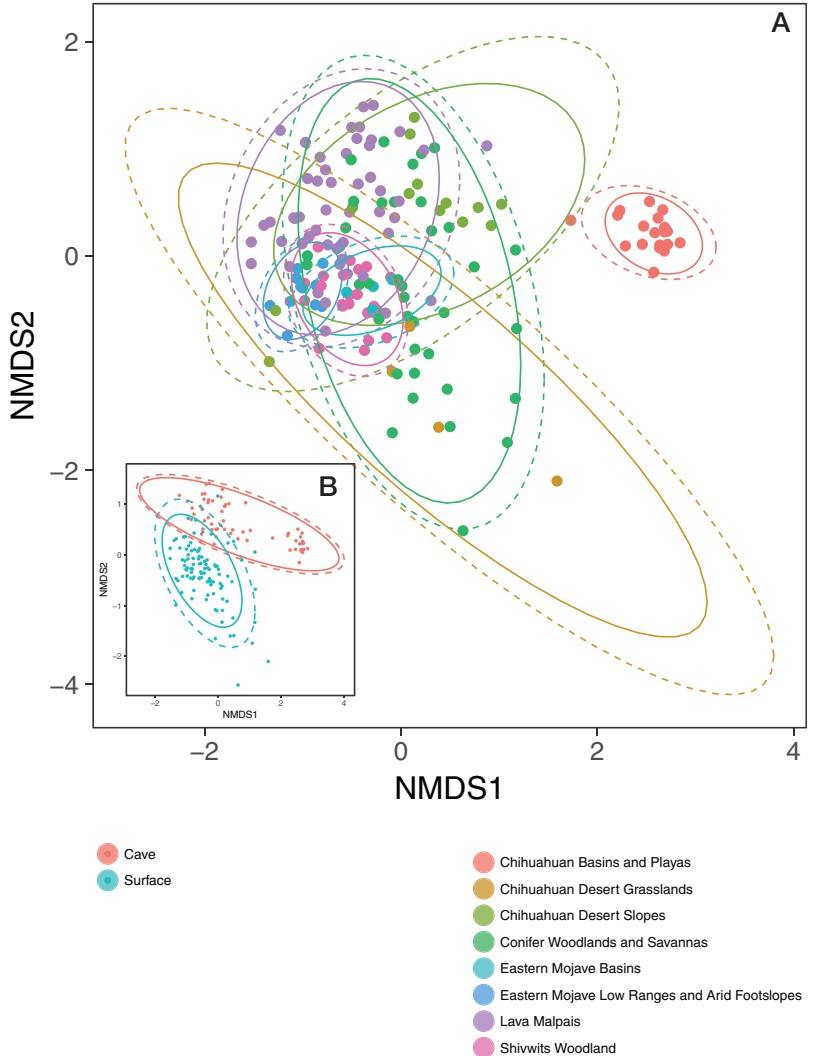

**Figure 4 Microbiomes cluster based on habitat and regions.** (A) NMDS colored by EPA ecoregion level IV. (B) (Inset) NMDS colored by cave-caught or surface-netted. Each point is a single sample. Points that are closer together are more similar in their microbiome composition. The ellipses surrounding each category of points are confidence ellipses. The dashed ellipses assume a multivariate normal distribution, while the solid ellipses assume a multivariate t-distribution.

at the 50% uncertainty interval (i.e., half the 50% intervals contain the true value). In the case of NMDS1, local habitat (sampling site) contributed −0.4 at Ft. Stanton Historic Section (from FS) to +0.3 at HGL-TOR (from Chihuahuan Basins and Playas). Feeding flight behavior contributed between −0.2 and +0.1. Bat species ranged from −0.7 to +0.7, while ecoregion was between −0.6 and +1.3. Cave-caught bats and surface-netted bats contributed 0. As latitude increased, community dissimilarity increased by −0.1.

All values are reported at the 50% uncertainty interval (i.e., half the 50% intervals contain the true value). Local habitat (sampling site) contributed −0.4 at Cerro Rendija to +0.6 at Rio Bonito Bridge (FS). Feeding flight behavior contributed 0. Bat species ranged from −0.1 to +0.2 and ecoregion was between −0.1 and 0. Cave-caught bats and

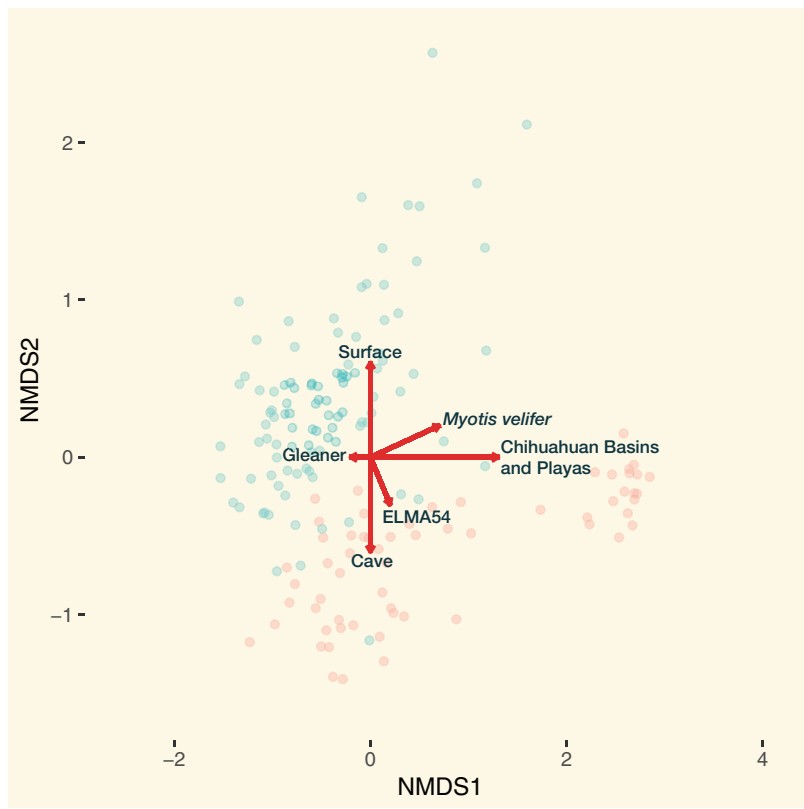

**Figure 5 Main drivers of bat skin bacterial community dissimilarity.** NMDS with samples colored by cave-caught or surface-netted. The arrows are the maximum variation in contributions to community dissimilarity derived from the Bayesian hierarchical models. Red circles of cave-caught bats and the blue circles are surface-netted bats.

surface-netted bats contributed +0.6 and −0.6, respectively. As latitude increased community dissimilarity increased by −0.1 on the NMDS2 axis.

The combined results of the NMDS1 and NMDS2 axis models are shown in Fig. 5. Using the Bayesian hierarchical model, bats from the Chihuahuan Basins and Playas had the strongest effect on community dissimilarity on bat skin/fur surfaces. Cave-caught or surface-netted were the second strongest drivers. *M. velifer* was also a driver, but bats from this species were primarily found in the Chihuahuan Basins and Playas region. El Malpais Cave 54 and gleaning feeding/flight behavior were smaller contributors to the bacterial community dissimilarity.

## BAT MICROBES ACROSS THE LANDSCAPE

Total bacterial richness on bats varies across latitude, ecoregion, cave-caught or surface-netted, and number of bats species present (Fig. 6). The Bayesian bootstrapped loess (nonparametric regression method) line was used to aid in visualizing the complex pattern emerging from bacterial richness.

A Bayesian hierarchical model was fit to explain the amount each predictor contributed to the bacterial richness on bats. All values are reported at the 50% uncertainty interval (i.e., half the 50% intervals contain the true value). The values reported here represent the

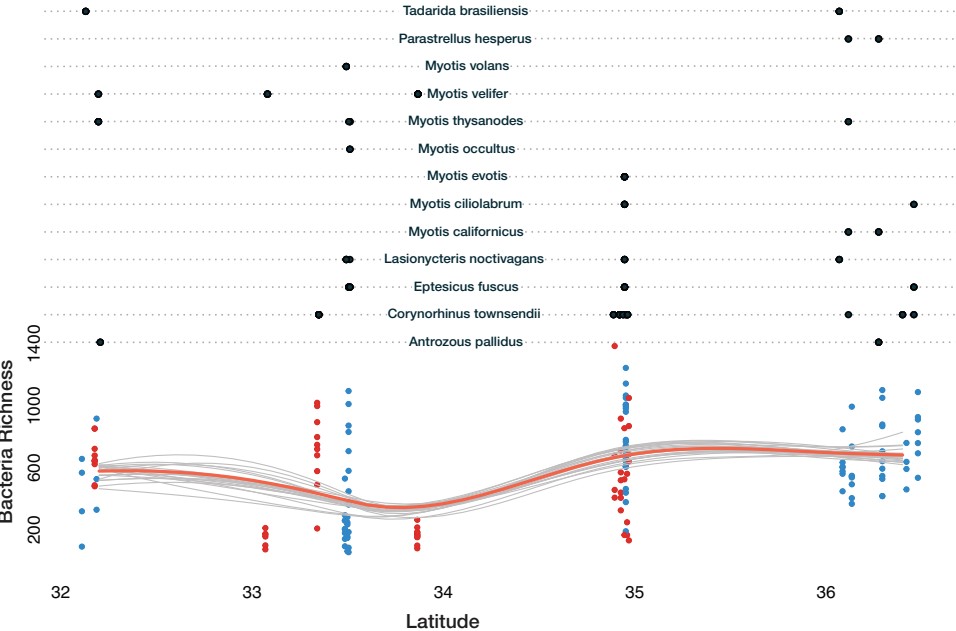

**Figure 6 Bacterial richness on bat skin increases with latitude in complex ways.** Bayesian bootstrapped loess line of bacteria richness on individual bats by changing latitude. The red line is the 95% uncertainty interval line and 20 additional gray lines show measures of uncertainty. This highlights the complicated interactions of bat species, latitude, habitat, ecoregion, and the impact on overall bacterial richness of the number of bat species present. Loess lines are for visualization only. The black dots in the upper portion represent the presence of the bats species at a given sampling location. The red dots are cave-caught bats and the blue dots are surface-netted bats.

contributions to overall bacterial richness on bats. A positive value of +57 for latitude, for example, means bacteria richness increases with increasing latitude. While a negative value, such as −329 at Rio Bonito Bridge indicates total bacteria richness decreases at this sampling site.

Local habitat (sampling site) contributed the greatest positive and most negative values to bacterial richness from two sites at FS: −329 at Rio Bonito Bridge and +298 at Rio Bonito. Local habitat also had the most variation out of all predictors. Feeding flight behavior contributed between −110 and +73 to bacterial richness. Bat species ranged from −26 to +25 and ecoregion was between −1.3 and 18.7. Cave-caught bats contributed 2, while surface-netted bats contributed −2. As latitude increased bacterial richness increases at +57.

Overall, bacterial richness shows a complex trend; however, there are underlying phylum level patterns that are difficult to evaluate. Figure 7 shows how total richness by select phyla varies across latitude, ecoregion, cave-caught or surface-netted, number of bats species present, and bat species presence/absence. Overall, at the phylum level, bacterial richness on bats increases with increasing latitude. However, latitude and habitat are cofounding factors and are not easily disentangled. Given the other supporting data presented local habitat is the stronger driver in this study. Actinobacteria and Nitrospirae

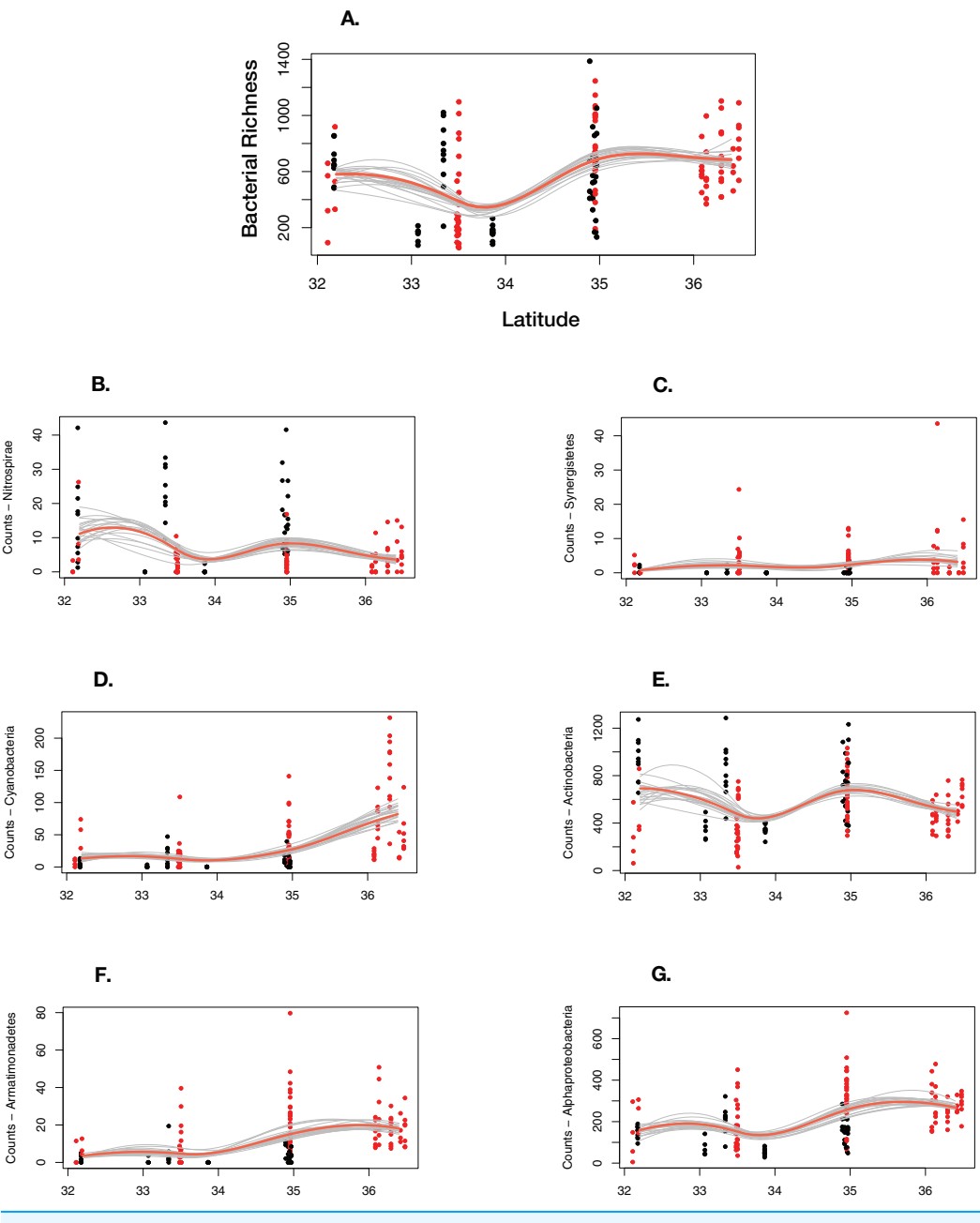

**Figure 7 Differential response to latitudinal gradient by bacteria phylum and class.** (A) Overall bacterial richness for each sample by latitude. (B) Nitrospirae. (C) Synergistetes. (D) Cyanobacteria. (E) Actinobacteria. (F) Armatimonadetes. (G) Alphaproteobacteria. Bayesian bootstrapped loess line of bacteria phylum richness on individual bats by changing latitude. The red line is the 95% uncertainty interval line and 20 additional gray lines show measures of uncertainty. The red dots are surface-netted bats and the black dots are cave-caught bats.

richness are higher at locations with more cave-caught bats and lower when there are more surface-netted bats. Synergistestes and Armatimonadetes are common soil bacteria and were expected to have higher richness in sites where there are more surface-netted bats. Cyanobacteria richness increases sharply with latitude and typically is higher in sites
that contain more surface-netted bats. Alphaproteobacteria (class) is common across most bat species and exhibits stronger latitudinal trends.

## DISCUSSION

### Overall bacterial diversity on bats

Actinobacteria was the most prevalent phylum on all bat species sampled, except for *T. brasiliensis*, where it was the second most prevalent after Firmicutes. This result is similar to the study by *Lemieux-Labonté et al. (2016)* that found Actinobacteria to be the largest component of the skin microbiota in both neotropical bat species studied at one location and the second largest, after Proteobacteria, at the second location. However, the study by *Lemieux-Labonté et al. (2016)* looked at only captive bats. *Cheng et al. (2015)* showed that skin microbiota can vary between captive and wild individuals of the same species. When we compare our study to other wild-caught bats, we find that bats in New Mexico and Arizona have a higher percentage of Actinobacteria and a lower percentage of Proteobacteria than those caught in Colorado, New York, and Virginia (*Avena et al., 2016*). While *Avena et al. (2016)* found that Proteobacteria comprise 63% of the skin microbiota of bats in their study, we found Proteobacteria to be on average only 24.6% of the microbiota. When the individual bat species are examined in this study, the relative percentage of Proteobacteria ranges from 8.15% (*T. brasiliensis*) to 39.98% (*M. volans*). This variation in the percentage of Actinobacteria and Proteobacteria may result from the differences between cave and surface sampling locations. In this study, Actinobacteria were found in a higher abundance on cave caught bats. Moreover, 37% of bats in this study were cave caught, in comparison to 16% in *Avena et al. (2016)*. Additionally, different hypervariable regions of the 16S rDNA gene were sequenced in the three previous bat microbiome studies. For example, both *Lemieux-Labonté et al. (2016)* and *Avena et al. (2016)* sequenced the V4 region, whereas we sequenced the V1–V2. The combination of different primers, as well as more cave-caught bats in this study, could account for some of the difference in the relative abundance of Actinobacteria. Furthermore, both of the previous external bat microbiome studies classified their sequences with the Greengenes database, while this study used the SILVA database to assign taxonomy. The different reference databases could account for some of the difference in abundance seen, especially in the unassigned taxa (*Yilmaz et al., 2013*). Specifically, *Avena et al. (2016)* reported between 7% and 59% of taxa at the class level could not be classified, depending on bat species, while this study reports 0–0.55% of taxa remain unclassified at the class level. Overall, the bats in this study appear to have a different skin microbiota compared to the two previous bat external microbiota studies, although some general patterns of high percentages of Actinobacteria and Proteobacteria hold true. Additionally, all three prior studies present evidence of a strong geographic influence on a bat's microbiome, similar to our study.

### Cave and surface habitats are primary drivers

Basic information on how roosting in a cave or flying on the surface affects a bat's skin microbiome is lacking. This is particularly important to understand when addressing novel wildlife diseases, such as WNS, that may alter naturally occurring microbiomes.

Because many bats can contract WNS while hibernating in caves, it is possible that the skin microbiome offers natural defenses against WNS for some bat species (*Hoyt et al., 2015*). Thus, the overall distribution of bacteria among phyla changing between cave-caught or surface-netted bats (Fig. S2) after a period of 6–8 h is important for bats vulnerable to WNS. Cave-caught bats had proportionally more Actinobacteria and Nitrospirae, while surface-netted bats had proportionally more Cyanobacteria, Firmicutes, and Synergistetes. Earlier studies in caves showed differences in community structure between surface soil and cave samples. This was seen in a carbonate cave speleothems in Arizona (*Ortiz et al., 2014*) where cave and surface soil samples were composed of different taxa. The photic and aphotic zone in samples from two caves in the Antarctic (*Tebo et al., 2015*) showed shifts in taxonomy with Cyanobacteria dropping off in the dark zone. Lava cave microbial mats in Lava Beds National Monument showed profound differences from their paired surface soil samples (*Lavoie et al., 2017*). In our current study, cave-caught and surface-netted bats were a main driver of bacterial community dissimilarity. While overall diversity indices were not different between cave-caught and surface-netted bat skin microbiota, which include active and dormant cells, the differential abundance and community dissimilarity provide evidence of taxonomic turnover between the two groups in a short period of time.

## Untangling confounding effects on the bat skin microbiota

From our study we found that geographic location, habitat type, species diversity, and likely behavior (Fig. 5) of the species will dictate bacterial diversity and community structure found on bats in the southwest. This is particularly evident when we examine the regional habitat relative to geography, local habitat, host species, and bat behavior of samples from the Chihuahuan Basins and Playas. Within this region the habitat was comprised of sparse, high desert grassland, with little topographic relief, which also contributed to driving the bacterial community dissimilarity (Fig. 5). This habitat was unlike the other habitat types occurring in other regions of our study that were much more diverse in vegetation and topography, such as the woodland/grassland/pine forest ecotone at ELMA. In addition to a near monotypic habitat type, bats from the Chihuahuan Basins and Playa region of New Mexico were sampled from below ground in two caves. Therefore, below ground sampling in these caves was a major driver in the bacterial community dissimilarity for these samples. From the same Chihuahuan Basin and Playas region, the low bacterial species diversity and *M. velifer* were strong drivers for bacterial community dissimilarity. We believe that *M. velifer*'s roosting behavior of multiple individuals grouped tightly against each other and against the cave wall/ceiling was a factor in driving bacterial community dissimilarity. This is particularly true, relative to other species sampled in this study, where individuals of different species were scattered along different locations of cave walls and in different caves.

Other factors that appear to be driving the bacterial community dissimilarity include qualities of local habitat below and above ground (where the bat was sampled). For example, although ELMA Cave 54 at El Malpais National Monument in western New Mexico is visually similar to nearby caves, this particular location floods

regularly, has few visible Actinobacteria colonies on the walls, is near a parking lot, and was accessible to the public at the time of sampling. It is likely that these factors, as well as others, can affect bacterial communities found on the skin/fur surfaces of bats. In addition to the aforementioned roosting behavior of bat species, such as noted for *M. velifer*, it appears that bat feeding-flight behavior is also a driver to the bacterial community dissimilarity. For example, *M. evotis* is a western species that is considered, at times, an insect gleaner and is capable of detecting and gleaning insect prey from substrates while in flight (*Faure & Barclay, 1994*; *Reduker, 1983*). The grassland/woodland/ pine forest ecotone of El Malpais National Monument provides suitable habitat for *M. evotis* to forage around vegetation and glean insects. In doing so, we believe that this feeding behavior provides opportunities for bats to obtain bacteria generally associated with plant material or surfaces of vegetation and the insects on or near them.

## CONCLUSION

Throughout this study, we found that the taxonomy of skin/fur bacteria found on bats caught in cave environments tend to be more homogeneous, compared to those found on bats captured on the surface. This can be seen in the cave-caught bats from across the landscape sharing the two dominant phyla of Actinobacteria and Nitrospirae. Both of these phyla are known to be prominent members of the cave microbial communities (*Lavoie et al., 2017*).

In light of these findings, there is an important implication that the skin/fur surfaces of bats from the West can be categorized into two distinct microbiota "worlds," depending on their capture. One world is comprised of a diverse spectrum of microbiota taxa that is generally found on surface-caught bats. The increased diversity of bacteria found on surface-caught bat is believed to be influenced by multiple biotic and abiotic variables that each species of bat encounters during their nightly flying bouts. The second world is comprised of microbiota that is collapsed (pruned taxonomy) and enriched in Actinobacteria and Nitrospirae that occur on cave-caught (roosting) bats. This reduced number of microbiota taxa encountered in cave environments is likely attributed to the reduced and stable number of biotic and abiotic variables present. The large volume of Actinobacteria taxa found in caves is of important note due to this phylum being prolific secondary metabolite producers. In some instances, some Actinobacteria can serve as possible biocontrols against wildlife diseases such as WNS. For example, Actinobacteria isolated from bats show inhibition of *P. destructans*, the causal fungal agent of WNS (*Hamm et al., 2017*). The decrease in taxonomic diversity in cave-caught bats suggests a possible mechanism by which *P. destructans* or other pathogenic fungus can gain a foothold (*Wargo & Hogan, 2006*).

Overall, our results shed new light on the skin/fur microbiota of southwestern bat species, including seven species with no previous microbiota studies. Furthermore, we provide new insight to which geographic, biotic, and abiotic factors influence the bacterial diversity patterns observed on different bat species. We have demonstrated that next generation sequencing at the landscape scale provides valuable information on bat skin microbiota. In particular, we have shown there are two worlds of external microbiota

found on bats, depending on whether the bat was cave-caught or surface-netted. Therefore, both management practices and biocontrol studies should focus on cave-caught and surface-netted bats with considerations towards local habitats as well as geographic region where these bats occur.

Future studies should focus on two areas: determining transient bacteria from permanent residents and determining active bacteria. The first study would require a large number of samples. Bats likely acquire their bacteria from the cave walls, air, trees, rock crevasses, and other area they opportunistically inhabit. The environmental samples would help to identify source and sink bacteria on bat skin and fur. There is a need to determine which bacteria are active participates on the bat skin and when they are active (in the cave or on the surface). A well-defined (single sample site) transcriptomics study would be invaluable.

## ACKNOWLEDGEMENTS

We thank the staff at El Malpais and Grand Canyon-Parashant National Monuments, Carlsbad Caverns National Park, Bureau of Land Management, and the Fort Stanton Cave Study Project. We thank Graham Walmsley for writing suggestions; Brennen Reece for graphic design and typographic help; and Ken of Kenneth Ingham Photography for the bat photo. Any use of trade, firm, or product names is for descriptive purposes only and does not imply endorsement by the U.S. Government.

### Funding

This work was supported by Colorado Plateau Cooperative Ecosystems Studies Unit (CPCESU)—Carlsbad, Caverns National Park (CAVE) Award #P14AC00793, UNM-101; Colorado Plateau Cooperative Ecosystems Studies Unit (CPCESU)—El Malpais National Monument (ELMA) Award #P14AC00588, UNM-99; Colorado Plateau Cooperative Ecosystems Studies Unit (CPCESU)—Grand Canyon Parashant National Monument (PARA) Award #P12AC10812, UNM-80; Fort Stanton Cave Study Project (FSCSP) and Bureau of Land Management Agreement No. 13-0484; Colorado Plateau Cooperative Ecosystems Studies Unit (CPCESU)—United States Geological Survey Award #G13AC00111; T&E, Inc. Award #TE-EAA-01222014; New Mexico Game & Fish Department Share with Wildlife Award #12516000000045; Western National Parks Association; National Park Service (El Malpais National Monument)—USGS cyclical funding Natural Resources Preservation Project-2013; National Park Service-NPS Reimbursable funding Agreement P14PG00266; United States Geological Survey—Salary funding. The funders had no role in study design, data collection and analysis, decision to publish, or preparation of the manuscript.

### Grant Disclosures

The following grant information was disclosed by the authors:
Colorado Plateau Cooperative Ecosystems Studies Unit (CPCESU)—Carlsbad, Caverns National Park (CAVE) Award: #P14AC00793, UNM-101.

Colorado Plateau Cooperative Ecosystems Studies Unit (CPCESU)—El Malpais National Monument (ELMA) Award: #P14AC00588, UNM-99.
Colorado Plateau Cooperative Ecosystems Studies Unit (CPCESU)—Grand Canyon Parashant National Monument (PARA) Award: #P12AC10812, UNM-80.
Fort Stanton Cave Study Project (FSCSP) and Bureau of Land Management Agreement No. 13-0484.
Colorado Plateau Cooperative Ecosystems Studies Unit (CPCESU)—United States Geological Survey Award: #G13AC00111.
T&E, Inc. Award: #TE-EAA-01222014.
New Mexico Game & Fish Department Share with Wildlife Award: #12516000000045.

## Competing Interests

Debbie Buecher is an employee of Buecher Biological Consulting.

## Author Contributions

- Ara S. Winter performed the experiments, analyzed the data, contributed reagents/materials/analysis tools, wrote the paper, prepared figures and/or tables, reviewed drafts of the paper.
- Jennifer J.M. Hathaway contributed reagents/materials/analysis tools, wrote the paper, reviewed drafts of the paper.
- Jason C. Kimble performed the experiments, analyzed the data, wrote the paper, reviewed drafts of the paper.
- Debbie C. Buecher conceived and designed the experiments, performed the experiments, contributed reagents/materials/analysis tools, wrote the paper, reviewed drafts of the paper.
- Ernest W. Valdez performed the experiments, contributed reagents/materials/analysis tools, wrote the paper, reviewed drafts of the paper.
- Andrea Porras-Alfaro wrote the paper, reviewed drafts of the paper.
- Jesse M. Young performed the experiments, analyzed the data.
- Kaitlyn J.H. Read conceived and designed the experiments.
- Diana E. Northup conceived and designed the experiments, performed the experiments, contributed reagents/materials/analysis tools, wrote the paper, reviewed drafts of the paper.

## Animal Ethics

The following information was supplied relating to ethical approvals (i.e., approving body and any reference numbers):

The field research and sample collecting as approved by: 2014 Arizona and New Mexico Game and Fish Department Scientific Collecting Permit (SP670210, SCI#3423, SCI#3350); National Park Service Scientific Collecting Permit (CAVE-2014-SCI-0012, ELMA-2013-SCI-0005, ELMA-2014-SCI-0001, PARA-2012-SCI-0003); Fort Collins Science Center Standard Operating Procedure (SOP) SOP#: 2013-01; and Institutional Animal Care and Use Committee (IACUC) Permit from the University of New Mexico (Protocol #15-101307-MC) and National Park Service (Protocol #IMR_ELMA.PARA.CAVE.SEAZ_Northup_Bats_2015.A2).

## Field Study Permissions

The following information was supplied relating to field study approvals (i.e., approving body and any reference numbers):

Field experiments were approved by the: 2014 Arizona and New Mexico Game and Fish Department Scientific Collecting Permit (SP670210, SCI#3423, SCI#3350); National Park Service Scientific Collecting Permit (CAVE-2014-SCI-0012, ELMA-2013-SCI-0005, ELMA-2014-SCI-0001, PARA-2012-SCI-0003); Fort Collins Science Center Standard Operating Procedure (SOP) SOP#: 2013-01.

## DNA Deposition

The following information was supplied regarding the deposition of DNA sequences:

DNA sequences are available at https://zenodo.org/record/50976.

## Data Availability

Biome files, QIIME mapping files, workflow, and R scripts are archived at https://zenodo.org/record/17577#. A Binder (http://mybinder.org/) ipython notebook with the full dataset used in this paper is available at: https://github.com/bioinfonm/microBat.

## Supplemental Information

Supplemental information for this article can be found online at http://dx.doi.org/10.7717/peerj.3944#supplemental-information.

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
