# Peer review of "Skin and fur bacterial diversity and community structure on American southwestern bats: effects of habitat, geography and bat traits"

_PeerJ, doi:10.7717/peerj.3944_

## Round 0.1 · original submission · Major Revisions

Both reviewers agree that the study is important even if the methods of analysis need polishing as well as a much better explanation. So even if the sampling and molecular analysis does not need to be repeated, the analysis of these data should be redone following the reviewers suggestions. As it stands now it cannot be compared with other studies, hence its scientific relevance becomes null.

Reviewer 1 ·

Basic reporting

This paper describes the bacterial communities found in the fur and skin of 14 species of insectivorous bats (in a total of 186 animals) in 5 caves in the USA states of Arizona and New Mexico. Bats were caught either directly in the caves or outside of them, from 2011 to 2014. Using 454 massive sequencing, the authors analyzed 16S rRNA.
The sampling appears to be good (but we are never informed of the total samples per bat species per cave, for instance), and in general, the molecular and statistical methods seem competent.
The main result is that, according to statistical analyses, cave- caught vs. surface-netted was the most important factor in determining the differences in the bacterial communities, and the second more important factor was the eco-region were the animals were sampled.
In relation to these two main findings, I have my most important concerns of the study.
1) Why do the bacterial communities associated to the bats change so much in a cave and outside? If they change so drastically, perhaps most of the bacteria the authors are finding are just transient, just acquired from the environment, but actually not living and interacting in the skin and hair of the bat, and perhaps they are not even active, since 16S rDNA does not distinguish if they are dormant or even dead.
It would be critical to design a method to actually evaluate the active and resident bacteria in a bat: the bacteria that are alive, interacting with other microorganisms, both bacteria and fungi, and with the other organisms in the skin (ectoparasites) and also organisms conducting a functional role in the external biome of the bat.
While we could devise molecular methods for disentangling this (to know which bacterial are the active vs. dormant or dead bacteria), such as analyzing the transcriptome of the bacterial RNA in the skins and hair of the bat (and should be discussed!), at least the authors could compare the bacterial communities that were found in a given bat species in a given time, both inside and outside the cave. These shared OTUs between the biomes inside and outside the cave may represent the true resident bacteria.

2) The sampling scheme is never clear. In particular, only 5 caves were analyzed, but in Figure 5 there are 8 environments. A table indicating caves, the external sites, sampling dates and bat species would help understand the final sampling design and to evaluate how well balanced is the study and which other possible analyses could be conducted with confidence. Also a clarification indicating if the bats in the caves are the same as the ones in the nets, or if the caves represent another population -- that may explain some of the results-- would be important.

Experimental design

Collecting Methods: Lemieux-Labonté et al. 2016, used as a negative control a humidified sterile swab, page 4, that sounds like a good idea, as they detected evidences of contamination, page 5. Apparently in the manuscript there were not similar controls; this cannot be changed if it is the case, but this point at least should be discussed.
Also, environmental samples of both the cave and the outside samples would have been interesting to check for the identity of the bacteria, instead of using the global or arbitrary data bases you used. Discuss.

Validity of the findings

I think the main findings are adequate, but the presentation of them should be improved and better interpreted. See the other sections for comments on the validity.

Additional comments

Abstract:
Lines 28: Briefly, explain how and why the microbiomes from day to night can be so different! At first glance, this does not seem to make any sense.
Line 31: Explain what is WNS.

Introduction:
In general, I feel that more emphasis should be given to the bacterial communities in the skin and hair in the Introduction.
The skin and hair of the bats should represent complex environments, as the bats are notorious because the high number of ectoparasites found, including fleas, different mites, and other arthropods, and their immune system seems to be very developed. Clearly the environment also includes fungi, as the one responsible of the white-nose disease.
The authors should make an effort to discuss better bacterial and the communities of other organisms from the skin (and hair in mammals or feathers in birds) in vertebrates.
An important reference that should be reviewed in the Introduction and more carefully discussed is the study of Lemieux-Labonté et al. 2016. PeerJ. The introduction of that paper cites several relevant references to understand the skin bacterial ecology that should be explained in the ms. Also, apparently this is the only similar analysis of culture free bacterial communities in bats, and for this reason it should be carefully described in the Introduction, and compared in the Discussion. The paper is indeed mentioned in the Discussion in line 299, but is not mentioned in the Introduction and the paper is missing in the References section.

Line 46: “in the Southwest”: Make it more general, so people outside North America will understand that you are talking about a specific part of the South of USA.

Line 49: I am not sure what “syntopic” means, explain.

Around line 66: Improve the review of the skin and hair communities in animals, in particular for mammals and bats. You mention the white-nose fungus, but as I explained above, many other organisms live there, and there are some interesting recent papers included and reviewed in Lemieux-Labonté et al. 2016.
Also, a critical issue is the functional role of the bacteria in this environment, as it can include transient (that may be interesting but irrelevant), and resident bacteria, and methods to disentangle both.

Lines 91 to 93: And if fungi communities are so important, Why not conduct a parallel study using ITS for the fungi? Not in this paper, but here you could discuss this (perhaps some bacteria defend from fungi in general) and you could use the DNA already extracted for this paper.

Methods:
As I mentioned above, the sampling is not clear, as there are 5 caves, but 8 eco-regions in Fig. 5.
A table stating the sampled caves, eco-regions, dates, and number of bats per species will help the reader to understand the total sampling, and how unbalanced was the sampling, and if it is possible to compare microbiomes of a bat species and a given date with samples in and outside the cave, to explore for the existence of resident and transient species.

Lines 111 to 122: As I mentioned above, some controls for contamination seem to be missing, and environmental samples would have been useful to interpret the data.
Also, how do you decide the number and species of bats in each cave and sampling? How many animals were sampled at each sampling date?

Lines 150 to 151 and 219 to 228: It is not clear if you rarefied or not the data, or rarefied in some analyses and not in other.
I am not sure if McMurdie and Holmes 2014 is actually right, but I am sure that using different sample sizes would bias the estimates of diversity.

Line 201: What is the meaning and interpretation of the VDS. I am not familiar at all with this statistic.

Results and Discussion:
In general, I do not like to make a joint section of Results and Discussion, as I feel the authors tend to confuse what they found and what the wanted to find. Separating both sections in clearer for the reader.
I strongly recommend to separate both sections, and to carefully rewrite the Discussion, making clearer comparisons with other studies of external microbiome in animals, in particular to Lemieux-Labonté et al. 2016 paper, as I mentioned above.

Fig. 1: It is very attractive, but make it more general: a) Mark the sampled caves. b) As these caves are more related to the southern populations, and as the national borders are meaningless for the biology, also include (North of?) Mexico (as Canada is included). This would make the map and the paper more useful and even more attractive.

Line 247: The number of reads among samples is very variable, 843 to 20,515 and thus I would be very careful of comparisons between samples if they are not rarefied. What was the average number of reads?

Lines 252: An easy to read and compare figure of the dominant bacterial groups in each sample, and perhaps in other categories, in particular of the bat species, would be important.
For instance, you could show this data in a simple pie figure, as in Figure 1 of Lemieux-Labonté, et al. 2016. If you use the same order and colors, this figure could be directly compared and discussed.
In contrast, your Figure S1 is completely useless. It is impossible to read or to make any sense of it.

Lines 255 to 264: The Random Forest results apparently are the main finding of the paper. I am not familiar with the method, but I think that if it is so important, these results should be shown in a figure or table, to give more emphasis, but also include the associate statistics to evaluate the probability or reliability of this analysis.
On the other hand, I am not so sure of the results of the Random Forest, given the values shown in Figure 7 for some other variables as soil PH, the values for some environments and some bat species, or sampling dates, that seem to be very important. I suggest to restructure the paper and give the Random Forest and the MDS analyses together, to try to disentangle what is happening and which are the more important variables that determinate the bacterial communities.

Lines 265 to 278: Above, in lines 252 to 254, there is a brief description of the communities. Put both sections together, to support the suggested above pie figure.

Figure 3 is interesting, but obscure, as the Y-axis is a complicated and non-intuitive variable (Log-relative normalized). Adding simple (additional?) pie figures or something like that may help understand the abundance of the bacterial groups. Also, explain better the estimate of the number in the Y axis, I guess a higher number means a higher number of that taxa, but it is not obvious.

Line 270: Supplementary figure S5 is cited here, but apparently the figures (or tables) S2 to S4 have not been cited yet. S5 figure is very complex, and I am sure there should be an easier way to visualize this information. Improve, please.

Lines 299: Extend and improve discussion of Lemieux-Labont et al. 2016. Also include reference.

Fig. 5: I do not see any gray line. May be just a problem of my computer, but it is a good idea to make sure the figure can be opened OK in any software.

Figure 6: As this analysis explains only 4% of the variance, I would send it to supplementary material, and just mention the statistic values in the text, lines.

Line 317 and Figure 7: What is the meaning of the VDS corrected richness? How you interpret this statistic?

Lines 342 to 359: While I think this analysis is interesting, I am not sure how reliable it is, as the comparison with the data sets are very heterogeneous: Barberán et al. 2015 seems to be for the complete continent, Newton et al. 2011 for all the available lakes in the world, and for the caves it seems very arbitrary, lines 213 to 218.
I am not confident at all of this analysis. It would have been better to take as controls environmental samples of each sampling locality. Please, consider sending this to supplementary material and carefully discuss its limitations.

Line 372: The “C” of cave is missing.

Line 375: Indicate the dominant phyla.

In general all supplementary materials will need a careful evaluation and edition, and a list of these materials was missing in the paper (and one of them I could not open). Some of the materials are very difficult to understand what they are or their meaning and the text is some case is too difficult or impossible to read, as was the case of S1.

Reviewer 2 ·

Basic reporting

Winter et al. analyze the external surface microbiota of 14 species of bats across the Southwestern United States. This is a relevant study that addresses important questions about the factors influencing bat microbiomes. Unfortunately, I consider that the manuscript has several structural and technical flaws that prevent it from its publication in PeerJ the way it is. At the end of this report I describe general comments and suggestions to correct and improve the manuscript. Should the authors address the corrections and improvements suggested, I will be happy to review a new version of the manuscript. Specifically, the article fails in explaining in a clear way their main findings. The manuscript needs a better structuring of the information. The introduction requires a better background on skin microbiota and some of the figures can be improved.

Experimental design

Overall, this study is correct in terms of the experimental design except for one major aspect that fails: my major concern is the decision of swabbing the entire skin and fur surfaces. Based on previous studies in other mammals, skin microbiota differs across skin regions so the decision of swabbing everything will obscure some important patterns. Also, the authors emphasize the importance of this study in the context of an emerging disease (WNS) however their swabbing technique neither replicates previous studies like Hoyt et al., 2015 nor it is focused on the regions infected by P. destructans (wings and uropatagium). Another important aspect, is that swabbing all external surfaces might also catch a high proportion of transient bacteria which are not specifically associated to bat skin, therefore making the interpretation of the results difficult. The authors need to clearly explain why they made this decision and how this would influence their results.

Validity of the findings

Overall, this study is correct in terms of the validity of the findings. However at the end I have several suggestions that will improve this section.

Additional comments

General comments
Overall, I consider that the methods, results and discussion are confusing due to the way the manuscript is structured. It is not clear which are the main findings and it is hard to follow the main line of the manuscript. For example, in the abstract it is clear that ecoregion and location are the main factors influencing “external surface” microbiota, however in the results (using different methods) it is mentioned that species, sampling site, seasonality, NPP, soil pH, Conifer Woodlands and Savannas, and Chihuahuan Basins and Playas are also influencing bacterial diversity. The authors need to determine which are the main findings and streamline their results and discussion. This includes, evaluating which methods explain better their results.

I have some concerns with some of the methods and analyses. Specifically, I don’t think the mantel test is strong enough to mention that geographic distance is playing a role. Even though the regression is significant the R value is extremely low. I would not include this test. Also, I am not convinced by the comparison of specific bacterial taxa with environmental sources. Bacterial communities in nature are highly diverse and therefore if the authors want to compare environmental microbes with the microbes obtained from bats they should have collected samples from the exact site where the bats were sampled. In my opinion, this is the correct way of comparing environmental microbes with potentially symbiotic ones and also to determine the proportion of transient bacteria present on the bat surfaces.

Since the authors analyzed 14 different species it would be interesting to analyze if there were any patterns associated to the bats habits. Do these bats eat the same food? Do they have similar life histories? What about a phylogenetic component?

Specific comments

Introduction

The authors need to expand the background on skin microbiota and their role to prevent emerging diseases in bats and in other non-human systems that have been studied with more detail.

Line 42. Eliminate “second to rodents at 2277 species”.

Lines 55-64. I am not sure how this information is relevant to this study, since all of these aspects are not analyzed later in the text.

Lines 77-78. You need to be more specific here, since this is the only study about antifungal bacteria in bats. Also, I think this study needs to be discussed later in the text in the context of your results.

Lines 79-85. Need more references of skin bacteria in bats such as Lemieux-Labonte et al. 2016.

Methods

The methods are a little confusing and they are disorganized. A better structure would help the reader follow the text better. For example, Sampling also includes molecular techniques like PCR. Normalization is mentioned twice. Diversity estimates are divided in different sections (alpha and beta diversity, NMDS plots etc) and they should be included in one.

Line 118. Explain what is Ringer’s solution

Line 151. Why did you use a depth of 1500? Please explain this sentence better.

Line 191. Eliminate the last sentence that starts with “NMDS…”

Line 203-209. There is no need to include the R script that you used here, unless you decide to include all the other scripts used for this study. If so, I would send all the scripts to supplementary figures.

Lines 219-234. Explain with more detail how DESeq2 and CSS work. Also, explain better how the data obtained with these methods was correlated with rarefied data. If these were highly correlated with the rarefied data, why not use rarefaction? Maybe I am missing something here since I am not familiar with DESeq2 and CSS.

Results and discussion

The results in general are difficult to follow. The authors describe patterns of diversity associated to cave caught and surface netted and then explain pattern associated to ecoregions and then go back to describing cave caught etc. I would reorganize the results in subsections that reflect the main findings.

Line 249: By looking at the supplementary figures, the authors need to explain why there was such a large proportion of unassigned reads.

Line 252-254: This sentence seems out of place. I think it should go later in the results.

Line 255: Change “The data were tested using a random forest model to see if the data…” to “A random forest model was used to see if the data…”

Line 256-259: You mentioned in the methods that a reasonable ratio of the estimated
generalization error compared to the baseline error should be two or greater, i.e. the random forests classifier does at least twice as well as random guessing for an unlabeled community. So why do you say the method was minimally successful if a ratio of 2 or greater seems enough? Please explain

Line 274-277: To determine the effect of ecoregions and habitat with random forest the authors should only include regions where both bats from caves and from surfaces were collected. Otherwise it is not possible to distinguish the effect of the ecoregion versus location.

Line 285: Why chloroplasts? Please explain.

Line 285-288: This needs to be explained better. I would think the opposite, that cave caught bats would be dominated by bacteria adapted to cave environments and that surface netted would have a higher diversity of bacteria because they are in contact with many environments, soil, air, plants etc.

Lines 289-299: There needs to be a stronger discussion section comparing with previous findings. The studies mentioned here were done in different species of bats with different habitats and the samples were obtained from different origins (guano, guts or ocular surfaces). There needs to be a more thorough explanation of why the authors find differences or similarities with these studies. Why did you not find differences among host species? were the samples taken differently that what Lemieux-Labonte et al. did?

Lines 300-301: The authors need to discuss the fact that transient bacteria may be present in their samples. How do you take this factor into account?

Lines 302-304: This was mentioned in the previous section too.

Lines 337-338: What are the implications of these differences? Overall, throughout the text the presence of specific groups of bacteria need to be explained better: What are these bacteria characterized by? What does it mean if a specific group is enriched in cave caught or in surface-netted bats etc?

Lines 339-341: I don’t think these sentences are relevant for the discussion

Conclusion

Lines 362: Why does comparing with one soil study (Ma et al., 2016) is relevant? Please explain. I would suggest to focus on specifying which are the factors that mostly contribute to explain your data instead of comparing with other unrelated studies.

Lines 363: “Internal” means “gut”?

Lines 372: What does “.Ave” mean?

Lines 374-387: I think there needs to be a clearer conclusion. What are the main findings? Which factors determine the microbiome in bats? Which factors will require more studies in order to determine their relevance?

Figures and tables

Figure 1: I don’t think this figure is relevant in the context of this study.

Figure 2: This maps need a scale. If you decide to keep the mantel test you could make a figure with the map and the graph together? This is just a suggestion.

Figure 7: To make the interpretation easier I would highlight with some color the variables associated to each of the NMDS and richness.

---

## Round 0.2 · Minor Revisions

The authors addressed all the problems and the reviewers are satisfied, however, many minor details are still pending and should be addressed before final Acceptance

Reviewer 1 ·

Basic reporting

For this new version of the manuscript the authors changed the figures and answered some of my concerns.

Nevertheless, there are still some issues, in particular about the figures that the authors will need to correct in the next version of the paper.

Experimental design

See below.

Validity of the findings

Somewhere in the Discussion or in the Conclusions, the limitation of the study should be clearly discussed in order to try to improve similar research in the future. From my original review I copy some of points that are still needed for this:
1) “It would be critical to design a method to actually evaluate the active and resident bacteria in a bat”, and explain how you could solve this.
2) “Lemieux-Labonté et al. 2016, used as a negative control a humidified sterile swab, page 4, that sounds like a good idea, as they detected evidences of contamination, page 5. Apparently in the manuscript there were not similar controls; this cannot be changed if it is the case, but this point at least should be discussed.”
3) “Also, environmental samples of both the cave and the outside samples would have been interesting to check for the identity of the bacteria, instead of using the global or arbitrary data bases you used”.

Additional comments

Line 137: It should say “Figure 1”, as the figures were changed.

Line 147: You should mention Figure 2a around here.

Line 161: Figure 2b should be cited here.

Line 197: Explain better. Apparently all the samples belonging to a bat species were removed. If this is the case, mention this bat species here.

Lines 207 and 294: Explain what is BLM.

Lines 276 to 286 need work. Move lines 276 and 277 to lines 195, were this explanation seems to belong.
And move lines 279 to 286 to Sampling section, around line 162.
Lines 286, remove an extra dot at the end.

Figure 3: Include the units in the figure: 0 to 60 of what? And check the species of the bats and include the name of the “M” species that is missing in the figure caption. Also, put in italics the Latin names in the caption.

Lines 360: Remove an extra dot at the end.

Figure 5: Indicate what is the meaning of the colors of the dots in the graph. Change bat name to “Myotis velifer”

Figure 6: Indicate the axis of the figure: latitude and richness, and also indicat the meaning of the two colors, red and blue of the dots.

Figure 7: Indicate the meaning of the colors of the dots.

Line 522: “likely behavior”? Maybe it is flying or feeding behavior, as in the graph. The figure should be 5, not 6. Correct.

Reviewer 2 ·

Basic reporting

I consider that the manuscript has improved from the last version. The introduction and the results are clearer now. However, I still consider the manuscript requires some improvements before being considered for publication. In particular, I consider that the authors didn´t properly address some of my concerns regarding the effect that transient bacteria may have on their results and also, they need to correct a few sections and figures. My comments are shown below (general comments to the author).

Experimental design

I have no further comments regarding the experimental design.

Validity of the findings

I have some comments here but I will present everything below(general comments to the author).

Additional comments

I don’t think the term “external microbiota” is the most appropriate term: First I think it´s too vague, second, if external means skin and internal means gut I would argue that actually the gut is also external since it´s not INSIDE the host. I would modify these terms (external and internal) and be more specific: skin and gut microbiota maybe?.

The authors expanded the methods section and explained why the sampled the fur in addition to the skin. However, I still think that this approach will be likely catching transient bacteria from the environment. In fact, there is a hole section in the discussion (lines 509-519) where the authors discuss how the differences they found between bats from different habitats may be related to changes in the environmental bacteria present in caves in contrast to surface environments. One interpretation of these results is that the sampling scheme is promoting the capture of transient bacteria and therefore the microbial communities on the skin are just a reflection of the environment. Therefore, I strongly think that the authors need to address the main caveat of the study, which is that they didn´t took environmental samples to compare with and therefore they can't discard the possibility of transient bacteria being sampled.

Specific comments

L42-44: I suggest adding more references on microbiomes in other animals (i.e amphibians hydra) in addition to the whale paper.

L122-124: Maybe rephrase this first question. Something like: does the daily routine of bats (spending time outside or inside a cave) influences their skin bacterial composition?

L197-200: This should be part of the section Sequence procesing.

L236-237: Phylum? I guess you mean Class? Alpha, beta and gammaproteobacteria are classes.

L262-265: I don’t understand the difference between the first and the second sentence.

L276-277. This section should be included in the introduction.

L279-286: This section should go in the “sampling section”.

L299-306: I would add this paragraph with the next section and eliminate the first sentence which is unnecessary.

L308: Change “Our study stands apart from culture-based studies and other next generation sequencing studies by focusing on the diversity of the external bacteria from...” to “Our study focused on describing the diversity of the external bacteria of 163 wild-caught bats from…”

L314-316: Be careful here: you are confusing classes with phyla. Choose either one but fix the mistake here and throughout the manuscript and figures.

L334. Change the title of this section to something like: The effect of habitat, ecoregion and species type on skin microbiota

L357-360: I suggest eliminating this paragraph. I don’t think is necessary.

L362: I don’t think you need so many subtitles for the results section. I suggest eliminating this one.

L387-388: Eliminate this first sentence and add this paragraph as a continuation of the previous paragraph. You can start L379 with: “A bayesian hierarchical model was fit to explain the amount each predictor contributed to the community dissimilarity on the NMDS1 and NMDS 2 axis. In the case of NMDS1…. In the case of NMDS 2…

L410: eliminate: (defined by QIIME’s alpha_diversity.py)

Figures 6: This graph needs more work: What are the axis? Explain the top part of the figure, what are the colors of the dots? The same with figure 7. They both need to be clearer.

L424: I am not familiar with these bayesian hierarchical models; thus, I really don't understand the values described below. I strongly suggest explaining this part more clearly. Specifically, the meaning of the values and the sign of the value need to be explained for a general audience: What does a negative or positive value mean? The larger the value the greater the effect?

L436-440: This section to me is telling the readers that habitat and latitude are confounding factors and that maybe is not latitude but habitat what is explaining these patterns. I would reconsider presenting these data as it is.

L487-498: I don’t think this paragraph of the discussion is relevant for this study. The comparison with other animal system at the bacterial phylum level is, in my opinion, not informative. The bacterial phyla that the authors describe here are pretty much present, not only on the skin of different organism but also on any environment.

Correct all the “cofounding” by “confounding”.

---

## Round 0.3 · accepted · Accept

This time the authors took great care in modifying the ms in order to reply to all the reviewers concerns. It is a nice paper and I think it got much better with all the process.